# Knowledge and Attitudes of Medical and Nursing Students in Greece Regarding Child Abuse and Neglect

**DOI:** 10.3390/children9121978

**Published:** 2022-12-16

**Authors:** Dionysia-Chara Pisimisi, Plouto-Antiopi Syrinoglou, Xenophon Sinopidis, Ageliki Karatza, Maria Lagadinou, Alexandra Soldatou, Anastasia Varvarigou, Sotirios Fouzas, Gabriel Dimitriou, Despoina Gkentzi

**Affiliations:** 1Department of Pediatrics, Medical School, University of Patras, 26504 Patras, Greece; 2Department of Nursing, University of Patras, 26504 Patras, Greece; 3Second Department of Paediatrics, School of Medicine, P. and A. Kyriakou Children’s Hospital, National and Kapodistrian University of Athens, 115-27 Athens, Greece

**Keywords:** child abuse, neglect, child maltreatment, CAN, knowledge, students, Greece

## Abstract

Data on the knowledge and attitudes of healthcare practitioners in training regarding child abuse and neglect (CAN) are scarce. The aim of this study was to investigate the knowledge and attitudes regarding CAN of medical and nursing students in Greek universities. We performed a questionnaire-based e-survey on a convenience sample of students and recruited 609 students (366 medical and 243 nursing). An unsatisfactory level of knowledge in the field was reported overall. Most of the students (92.2%) were aware of their future responsibility to protect vulnerable children and report suspected cases of CAN; at the same time, they were willing to obtain further education. Based on the above, appropriate training in the undergraduate curriculum should be developed in order to strengthen future healthcare practitioners and boost their confidence in dealing with suspected cases of CAN and protect children’s welfare.

## 1. Introduction

Violence is a worldwide problem that affects all countries and communities [1]. According to the World Health Organization (WHO), “*violence against children includes all forms of violence against people under the age of 18 years, whether perpetrated by parents or other caregivers, peers, romantic partners, or strangers*” [2]. Child abuse and neglect (CAN) is one of the six main types of interpersonal violence against children. As stated by the WHO, CAN involves inflicting physical, sexual, and psychological/emotional violence and the neglect of infants, children, and adolescents by parents, caregivers, and other authority figures; most often these occur at home but also can occur in other settings such as schools and orphanages [2]. Child maltreatment or CAN can be divided into the following main types: neglect, physical abuse, psychological abuse, and sexual abuse. It should be highlighted that a particular form of child maltreatment known as Munchausen syndrome by proxy (also known as medical child abuse or factitious disorder imposed on another) is a condition that involves characteristics of physical abuse, psychological maltreatment, and medical neglect [3,4]. In another form of Munchausen syndrome by proxy, a relatively rare disorder, the primary caregiver subjects the child to repeated medical interventions based on a false or induced history [5]. Munchausen syndrome by proxy was given the T74.8 code in the International Classification of Diseases, Tenth Revision (in which T74 includes all known forms of child maltreatment) [6].

According to “Smile of the Child”, a voluntary, non-profit child welfare organization based in Greece, the number of reports made for CAN in the first half of 2022 were 557 [7]. These reports appear to have involved 964 children. Based on these data, violence against children aged 0–12 years was the most frequent type. In addition, the most prevalent form of child maltreatment was neglect: 483 children were noted as having suffered from this type of abuse followed by 399 reports of physical abuse. Furthermore, when comparing these outcomes with those of the first half of 2021, the reporting of child sexual abuse and exploitation remained extremely low, which indicated that these cases might still remain unreported. In addition, it was noteworthy that in 486 children, a person of the immediate family environment was the abuser [7]. It is crucial to pinpoint that these statistics do not reflect the actual magnitude of CAN as a complex social issue because it is believed that the reported CAN cases constitute only a small percentage of the children actually suffering from CAN [8].

Healthcare professionals, especially those working with children, play a pivotal role in addressing this complex pathology because they are the first professionals that come into contact with the severe end of the spectrum, especially during infancy. This is why they are required to be familiar with the physical and behavioral signs in children and caregivers that are indicative of CAN. They should also be aware of implementing the proper protocols and, most importantly, reporting to the relevant services. It is worth mentioning at this point that other professionals such as teachers or police officers do come in contact with suspected cases of CAN in the community. The latter highlights that the burden of the responsibility to recognise signs of CAN is distributed amongst a wide spectrum of professionals because cases of CAN do not always end up admitted in the hospital. Moreover, it is of paramount importance for every citizen to be familiar with the process of reporting any suspicion of abuse to the police or child-protection authorities.

Although identification and management of CAN are skills that health professionals are required to possess, studies have shown that there is a gap in knowledge that is possibly due to inadequate training [9,10]. At the same time, little research has been performed on the knowledge and attitudes of healthcare practitioners regarding training [9,10]. Therefore, the aim of this study was to investigate the knowledge and attitudes regarding CAN of medical and nursing students in Greek public universities with a view toward designing educational strategies that focus on developing their skills in the identification and management of CAN during their professional life and intervening in a timely manner to potentially save a child’s life.

## 2. Materials and Methods

We performed an electronic cross-sectional survey by inviting via email the students of all the faculties of medicine and nursing in Greece during a period of 4 months (19 October 2021–16 February 2022) using a questionnaire developed by our research team and piloted in a small number of students prior to the beginning of the study. All potential participants were informed in detail about the study aims as well as the data confidentiality. Students who agreed to take part in the study were asked to complete the anonymous online questionnaire regarding their knowledge of CAN. The questionnaire was written in Greek and consisted of 4 parts. Part A included four questions on demographics (gender, university, faculty, and year of study). Part B included 16 questions on their knowledge and attitudes regarding CAN. Part C included six questions regarding the education they had received from their faculty’s curriculum or elsewhere. Part D included three questions on their clinical and non-clinical work with children. A total of 13 answers were given on a 5-point Likert scale (“extremely well”, “well”, “moderately”, “little”, or “not at all”; or “very much”, “much”, “moderately”, “little”, or “not at all”).

Statistics were determined using the IBM SPSS software version 27 (IBM Corp., Armonk, NY, USA). The level of significance was set to 0.05 for all analyses. Descriptive statistics were used to summarize the data on demographic characteristics and knowledge and attitudes regarding CAN. The chi-squared test was used to determine any significant associations between the demographic characteristics and a student’s knowledge and attitude. Uncertain responses were scored as incorrect. During the statistical analysis, the answers “moderately”, “little”, and “not at all” were grouped as “below average”, whereas the rest were grouped as “above average”.

Ethical approval was obtained from the University of Patras Ethical and Research Committee (application ID: 8409, 19 September 2021).

## 3. Results

### 3.1. Demographic Characteristics

Out of a total of approximately 12,500 students in the faculties of medicine and nursing at public universities in Greece, where our online questionnaire was distributed, 609 responded (4,87% response rate). The entire pool of students received the link to participate in the study; however, very few responded. Table 1 shows the demographics of our studied population. Of note, the majority of our study participants were women (75%) (Table 1).

### 3.2. Students’ General Knowledge of CAN

Table 2 shows the self-reported knowledge of students on the types of CAN, its most prevalent type worldwide, and who they believe might be responsible for it. Apart from these, during the assessment of the responses, we noticed that a great number of students were not able to name psychological abuse using its appropriate term; instead, the responders used other terms such as emotional or verbal abuse or even specific examples of it in an attempt to explain what they knew. Since the respondents had the opportunity to identify more than one perpetrator of CAN, the total percentage exceeded 100% (Table 2).

### 3.3. Students’ Education on CAN

Table 3 describes the characteristics of the education that students had received through their faculty curriculum as well as the type of exposure they had as part of their training. In addition, the table shows how many of the respondents had previous experience in working with children (whether in a hospital setting or not) and how many of them had already come into professional contact with abused or neglected youth (Table 3). The term working experience referred to whether students had any kind of professional contact with children. Clinical experience referred to students that were engaging in or had completed their clinical practice on the pediatric ward round (such as final-year students that were working as house officers or nursing students who were engaging in their practice on the ward), whereas non-clinical experience referred to students that had worked with children outside the healthcare setting in situations such as babysitting, camping activities, etc.

### 3.4. Effect of Demographic Characteristics on Knowledge and Attitudes on CAN

The correlations between the demographic characteristics, knowledge, and attitudes regarding CAN are shown in Table 4. Of note, no association was found with the university that the study participants attended. Various other associations were demonstrated between knowledge, attitudes, and demographics.

## 4. Discussion

This was the first study performed on the knowledge and attitudes of medical and nursing students in Greece regarding CAN. The response rate was 4.9%, which was small yet rather satisfactory when taking into consideration the large number of medical and nursing students and the difficulties of recruiting them via electronic communication (the study was conducted during the COVID-19 pandemic). In addition, the small participation rate in our survey could be indicative of a possible unwillingness to deal with CAN in the future. According to a study performed in Australia, there was a significant likelihood of medical professionals failing to report suspicious child abuse and neglect cases due to numerous reasons such as a lack of specific indicators of abuse, uncertainty about the reporting requirements and procedures, knowledge of the family, and the perception that the abuse was a single incident. [11]. According to the results of our study, students in the faculties of medicine and nursing in Greece had an unsatisfactory level of knowledge regarding CAN. Concerning the forms of CAN, it came as a surprise that only 26% of medical and 19.8% of nursing students mentioned neglect as a possible form, which rendered it second to last after Munchausen syndrome by proxy. Similarly, only 4.9% of medical and 1.6% of nursing students believed that neglect is the most prevalent form of CAN. This can be explained by the lack of accuracy of the definition of neglect (failure of the parent to provide the necessities for the proper development of their child), which may have been due to a wide range of reasons such as cultural and traditional perceptions. This may be why students were not able to recognize neglect as such a widespread form of CAN. [1] These were quite alarming outcomes when considering that half of the reports of CAN in Greece by the “Smile of the Child” organization regard child neglect [7].

About half of the students (54.7%) acknowledged the possibility of specific characteristics of the child (such as age, sex, the existence of disabilities, etc.) affecting the risk of CAN, while approximately one-fourth of them (24.1%) falsely believed that specific characteristics of the parents/guardians (such as age, mental illness, substance use, etc.) did not affect their probability of inflicting CAN. Apart from this, our results concerning the students’ self-reported knowledge on suspected indications of CAN were quite astonishing because the majority of the students admitted to being not aware enough of the physical indications (64.4%), child’s behavior (73.4%), and parents’/guardians’ behavior (80.6%) associated with the increased suspicion of CAN. In a previous study performed in Greece, healthcare practitioners (the majority consisting of pediatricians, pediatric trainees, and nurses) reported inadequate knowledge of the factors that increase the risk of CAN, the physical signs, and the behavioral indicators [9]. In a study performed in the Republic of Korea, medical students were asked to participate in interactive clinical scenarios, some of which concerned victims of domestic violence, which is equally severe to CAN; lower performance scores were recorded in the domestic violence scenarios, especially regarding the students’ interpersonal skills and the organization of the interview [12]. In addition to the above, 9 out of 10 students (90.6%) presented average to non-existent knowledge of the clinical tools that are helpful in the assessment of suspected CAN, while 9 out of 10 students (89.7%) also reported limited knowledge of the procedures following the report of a suspicious case. Moreover, 90.6% of our participants were not aware of the services responsible for the post-report management of a suspicious case, which was in alignment with the beliefs of health practitioners in other areas, some of whom had negative experiences with child-protection services [13].

More than 80% of the students recognized parents and relatives of the child (either biologically related or not) as possible perpetrators of CAN. According to the “Smile of the Child” organization, during the first semester of 2022, 464 out of 545 reports included one or both parents as the perpetrator of CAN, while the *Child Maltreatment Report 2020* issued by the U.S. Department of Health and Human Services indicated that 77.2% of perpetrators were the parents of the victim of CAN, 6.6% were a relative other than a parent, and 4.2% had multiple relationships with the victim. Apart from that, 3.8% of the perpetrators had a different type of relationship with the victim [14].

According to our findings, the education provided by the medical and nursing curriculum was considered insufficient by the students themselves because only 19.7% of the participants reported having a chapter that covered CAN in their curriculum, while 27.3% reported that a professor took the initiative to discuss CAN. Similar findings can be seen in studies in China, the USA, and Saudi Arabia [15,16]. It is of interest to note that students in clinical years of medical and nursing studies reported a lack of confidence in handling a suspected case of CAN. This decrease in the self-esteem of the clinical-year students might have been due to their awareness of deficient knowledge, whereas preclinical-year students might not have been aware of their yet-incomplete capabilities and might have underestimated the complexity of CAN. Still, 80.8% of all respondents reported feeling moderately prepared or less to handle a suspected case of CAN in their future professional life. A similar lack of confidence was shown in studies on healthcare practitioners from Greece and other countries [9,10,12]. However, even in the case that they have been educated, it is still possible to feel rather unprepared, as shown by a Saudi Arabian study on the knowledge and attitudes of medical and dental students and interns regarding CAN. According to this study, there was no significant correlation between previous training on CAN and the level of knowledge of the subjects [17]. This implied that there is a great need for a more structured systematic approach to the training of future healthcare professionals in this field, possibly similar to the one described in a publication concerning train-the-trainer and workshop modules performed in Greece that targeted the improvement of the knowledge of physicians and other healthcare professionals regarding child physical abuse [18].

The majority (92.8%) of the study participants recognized the increased responsibility that falls on the shoulders of healthcare practitioners; this was especially expressed by the medical students. According to Greek law, certain instances/variations of bodily harm (of a differentiated intensity) committed either against a minor or in front of a minor by their parents or guardians are criminalized along with sexual relations/intercourse with minors below the legal age of consent [19]. Furthermore, the responsibility of healthcare practitioners who suspect CAN is not vaguely dispersed amongst the various healthcare professionals involved. Each one has an individual responsibility to act for the child’s welfare. The appropriate course of action when there is suspicion of CAN is the accurate reporting to the organizations responsible for that purpose or the police. This reporting does not constitute a formal accusation but a preemptive approach for gathering further information and performing the appropriate evaluation [20]. According to the United Nations Convention on the Rights of the Child, by which Greece is also bound, the States Parties shall *“recognize the right of the child to the enjoyment of the highest attainable standard of health and to facilities for the treatment of illness and rehabilitation of health”* [21]. In addition, the States Parties are required to provide all the appropriate measures for the protection of the child from every form of physical and psychological abuse, neglect, and maltreatment (including sexual violence) while the child is in the custody of their parent(s), legal representative, or any other person to whom they are entrusted [21]. This article included healthcare professionals as state workers who were required to have the child’s best interests as a priority.

Greece, as a Council of Europe (CoE) Member State, is bound by the European Convention on Human Rights (ECHR) and the 2007 CoE Lanzarote Convention on the Protection of Children against Sexual Exploitation and Sexual Abuse (CETS No 201) [22]. While the former (i.e., the ECHR) is a more generic international treaty, its monitoring body (the European Court of Human Rights) has a rich jurisprudence with regard to CAN in various circumstances [22]. On the other hand, the Lanzarote Convention specifically criminalizes any sexual activity with a minor (a child below the legal age of consent) under any circumstances (along with child prostitution and pornography) and imposes on the States Parties the obligation to adopt all necessary preventive measures against such phenomena (including—among others—the training and educating of children as well as people who are employed to work with children) [23]. Since Greece has signed and ratified both of these international instruments, according to the relevant constitutional requirements, the provisions of both conventions are integrated within the Greek legal order and supersede any other national legislation. Lastly, it is encouraging to see that 96.6% of students who received no form of education on CAN were willing to be educated on this issue. This intention was not correlated with the faculty to which the students belonged, their gender, or the year of their studies. A similar need was shown in other studies as well [16,17,24,25,26,27,28].

Another finding of our study that is worthy of comment was that the years of studies did not seem to significantly affect the level of knowledge. The latter was a worrying finding because one would expect that students in the final stages of their training would be more aware of the issue. However, we did not demonstrate such a correlation. Therefore, faculties ought to take this into account and amend their curricula to the extent that each healthcare professional in training obtains a basic understanding of CAN that will enable safe practices for children in their future. In most faculties there is no separate curriculum on CAN, but such training could be incorporated into the subjects of pediatric or forensic medicine or both. Another suggestion is to adopt a stepwise approach, meaning that training on CAN could be offered both in the early stages of training as well as in the final years. In addition, part of this structured training should build the understanding that all healthcare professionals, irrespectively of their seniority and especially those that deal with children, should recognize early signs of CAN and act appropriately. Of note, recent efforts have been made in Greece to increase the knowledge of medical students with the aid of novel methods such as role playing and the use of manikins [29].

This study, as a pragmatic one, was not without limitations. First of all, it was conducted using an online self-reporting questionnaire in which we had no direct contact with the participants; this might have discouraged participants from expressing any questions regarding the content of the survey. As we used convenience sampling to acquire these responses, it is possible that students with increased knowledge were more willing to respond. At the same time, it was not possible to acquire an equal response rate from each university and faculty, which can be partly attributed to the electronic nature of our questionnaire. In addition, we distributed the questionnaire to the entire pool of medicine and nursing students at public universities in Greece; even though the total number of responses collected was high, our response rate remained low. Another study limitation was that the training varied among faculties, which therefore affected the level of knowledge amongst the students. Moreover, more medical than nursing students participated in the survey because the completion of the questionnaire was done on a voluntarily basis. In addition, although our results may be representative of Greece, we cannot generalize them to other countries due to the different structures of educational systems and CAN reporting services. Finally, since these limitations exist, it would be judicious to conduct further research to better specify the level of knowledge inadequacy and thus develop the appropriate course of action. Among the aspects to be researched, we recommend including the correlation between the impact of CAN among students and their knowledge and attitudes regarding CAN. Of note in a previous Greek study, 1 in 5 medical students admitted that they were victims of sexual abuse during childhood and that this experience impacted their attitudes toward child sexual abuse [30].

## 5. Conclusions

In conclusion, our study outcomes supported our initial hypothesis that because there are no properly structured educational programs in the faculties of medicine and nursing in Greece for CAN students, students would report an unsatisfactory level of knowledge in the field. The majority of students were aware of their future responsibility to protect vulnerable children and report suspected cases of CAN; at the same time, they were willing to obtain further education. Based on the above, appropriate training during the undergraduate curriculum should be developed more systematically in order to strengthen future healthcare practitioners and boost their confidence in dealing with suspected cases of CAN and protect children’s welfare.

## Figures and Tables

**Table 1 children-09-01978-t001:** Demographic characteristics of the study population *(n* = 609).

Demographic Characteristics	Number (Percentage)		*p*-Value (*t*-Test)
	Medicine	Nursing	
Total number of students	366 (60%)	243(40%)	
Gender			
Male	118 (32.2%)	33 (13.6%)	<0.001
Female	243 (66.4%)	209 (86%)	<0.001
Other	5 (1.4%)	1 (0.4%)	NS (0.194)
Year of studies			
Preclinical years	136 (37.2%)	78 (32.1%)	NS (0.198)
Clinical years	230 (62.8%)	165 (67.9%)	NS (0.198)

**Table 2 children-09-01978-t002:** General knowledge of the study population regarding CAN (*n* = 609).

General Knowledge	Number (Percentage)		*p*-Value (*t*-Test)
	Medicine	Nursing	
Total number of students	366 (60%)	243 (40%)	
Types of CAN			
Physical	343 (93.7%)	213 (87.7%)	0.014
Psychological	353 (96.4%)	219 (90.1%)	0.003
Neglect	95 (26%)	48 (19.8%)	NS (0.072)
Sexual	287 (78.4%)	153 (63%)	<0.001
Munchausen by proxy	2 (0.5%)	0 (0%)	NS (0.158)
Other	55 (15%)	24 (9.9%)	NS (0.055)
Invalid	39 (10.7%)	39 (16%)	NS (0.06)
Most prevalent type			
Physical	108 (29.5%)	71 (29.2%)	NS (0.939)
Psychological	199 (54.4%)	121 (49.8%)	NS (0.269)
Neglect	18 (4.9%)	4 (1.6%)	0.019
Sexual	18 (4.9%)	28 (11.5%)	0.005
Munchausen by proxy	0 (0%)	0 (0%)	-
Other	9 (2.5%)	7 (2.9%)	NS (0.751)
Invalid	14 (3.8%)	12 (4.9%)	NS (0.507)
Variations of Psychological Abuse			
Verbal	257 (70.2%)	155 (63.8%)	NS (0.1)
Emotional	77 (21%)	44 (18.1%)	NS (0.367)
Mental	25 (6.8%)	20 (8.2%)	NS (0.519)
Intellectual	3 (0.8%)	4 (1.6%)	NS (0.35)
Other	13 (3.6%)	0 (0%)	<0.001
Possible perpetrators			
Biological parents	353 (96.4%)	220 (90.5%)	0.005
Adoptive parents	336 (91.8%)	201 (82.7%)	0.001
Guardians	327 (89.3%)	197 (81.1%)	0.006
Relatives besides parents	345 (94.3%)	195 (80.2%)	<0.001
Non-relatives familiar to the child	334 (91.3%)	190 (78.2%)	<0.001
Non-relatives not familiar to the child	301 (82.2%)	170 (70%)	<0.001

**Table 3 children-09-01978-t003:** Characteristics of students’ education (*n* = 609).

Characteristics of Students’ Education	Number (Percentage)		*p*-Value (*t*-Test)
	Medicine	Nursing	
Total number of students	366 (60%)	243 (40%)	
Chapter in the curriculum	56 (15.3%)	64 (26.3%)	0.001
Professor’s initiative	96 (26.2%)	70 (28.8%)	NS (0.485)
Education outside of the faculty (n = 378)	207 (56.6%)	171 (70.4%)	<0.001
Conference attendance	72 (35%)	55 (32.5%)	NS (0.593)
Workshops	26 (12.6%)	15 (8.9%)	NS (0.233)
Internet	147 (71.4%)	134 (79.3%)	NS (0.101)
Personal initiative	129 (62.6%)	97 (57.4%)	NS (0.272)
Other	11 (5.3%)	13 (7.7%)	NS (0.365)
Working experience with children (n = 371)	231 (63.1%)	140 (57.6%)	NS (0.176)
Clinical	112 (48.5%)	55 (39.6%)	NS (0.083)
Non-clinical	165 (71.4%)	111 (79.9%)	NS (0.085)
Working experience with abused children (of those mentioned above)	67 (29%)	39 (28.1%)	NS (0.813)

**Table 4 children-09-01978-t004:** Association of demographic characteristics with knowledge and attitudes regarding CAN.

			*p*-Value (Chi-Squared Test)		
Statement	Correct/ Acceptable	Incorrect/ Not Acceptable	Faculty	Gender	Year of Studies
Specific characteristics of the child affect their probability of suffering CAN	54.7%	45.3%	NS	0.021 Incorrect = other	NS
Specific characteristics of the parent/guardian affect their probability of causing CAN	75.9%	24.1%	NS	NS	NS
How well do you know the risk factors concerning CAN?	29.89%	70.11%	0.025 Not acceptable = medicine	NS	NS
How well do you know the physical indications associated with CAN?	35.6%	64.4%	<0.001 Not acceptable = medicine	NS	NS
How well do you know the child’s behaviors that are suspicious/indicative of CAN?	26.6%	73.4%	<0.001 Not acceptable = medicine	NS	NS
How well do you know the parent’s/guardian’s behaviors suspicious/indicative of CAN?	19.4%	80.6%	NS	NS	NS
Is there a chapter on CAN in your faculty’s curriculum?	19.7%	80.3%	<0.001 Not acceptable = medicine	NS	<0.001 Not acceptable = preclinical years
Have your professors taken the initiative to talk to you about CAN?	27.3%	72.7%	NS	NS	<0.001 Not acceptable = preclinical years
Have you received any form of education on CAN outside your faculty?	61.6%	38.4%	<0.001 Not acceptable = medicine	<0.001 Not acceptable = male	NS
If you have not received education on CAN, would you like to do so? (n = 232)	96.6%	3.4%	NS	NS	NS
How well do you know the clinical tools helpful in the detection of CAN?	9.4%	90.6%	NS	0.018 Not acceptable = female	NS
How well do you know the procedures following the report of a case suspicious of CAN?	10.3%	89.7%	0.016 Not acceptable = medicine	NS	NS
How adequate do you believe the services responsible for the management of a suspicious case, post-report, are?	9.4%	90.6%	0.004 Not acceptable = medicine	0.021 Not acceptable = other	NS
How much responsibility to report suspicious cases do you believe falls on the shoulders of healthcare practitioners?	92.8%	7.2%	0.017 Not acceptable = nursing	NS	NS
How prepared do you feel to handle a case suspicious of CAN?	19.21%	80.79%	0.018 Not acceptable = medicine	NS	<0.001 Not acceptable = clinical years

## Data Availability

Additional study data are available upon request.

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
