# Peer review of "Knowledge and Attitudes of Medical and Nursing Students in Greece Regarding Child Abuse and Neglect"

_children, 2022, doi:10.3390/children9121978_

Round 1

Reviewer 1 Report

Before getting to my overall impression I have a few specific comments to make. 

First, I'm puzzled why Munchausen's by Proxy is singled out as a specific category of abuse. It is generally accepted that it is an extremely rare condition and difficult to establish even when one finds it. I've read the original research by David Southall and Sir Roy Meadow, and while I'm sympathetic, one cannot avoid the political firestorms their various careers have created. I don't see how this is helpful to the present paper, and it could be harmful insofar as it could lead to sterile arguments about whether Munchausen's actually exists.  It certainly distracts from an important finding here, namely that while neglect is the most common form of abuse, it is the least likely to be recognized and understood. 

Now that I think about it, it would have been more useful to separate out Sudden Unexplained Death (or SIDS as was) since this is relatively common and highly problematic from a diagnostic point of view.

Second, on page two the claim is made that health care professionals are the 'first to have contact with CAN'. This may be true for infants, but surely, for school age children, the more likely 'first' professionals are teachers? Moreover, would not one expect that police officers would be the first to see abuse connected with domestic violence? I think if this claim is to be made, then it needs to be substantiated. This also implicates the point early in the discussion that medical professionals may be reluctant to participate in child abuse investigations. 

The Questionnaire: A major problem with child abuse reports is the nature of the requirements to report. Neither in the methods, nor in the subsequent discussion section, is it clear what gets reported, by whom, and who does the follow-up work and generates the statistics. My jurisdiction is simple; it is mandatory for every citizen to report any suspicion of abuse to the police or child protection authority. It's not perfect, but it does have the virtue of clarity. The only exception is Solicitor/Client privilege. This doesn't seem to be the case for Greece, but it's not clear. Further, it seems the definition of abuse - or at least harm - is determined by the criminal code rather than by civil law as it is elsewhere. These matters need to be clarified since, at least in my jurisdiction, doctors are notorious for wanting to avoid the courts. 

In passing I note there is a page numbering problem that needs to be addressed.

Discussion: The first paragraph of the Discussion section states "In addition, the small participation rate may indicate that future healthcare professionals might be unwilling to deal with CAN in the future."  There is support for this view, but it's not necessarily in the academic literature. Rather, you will find it in the grainy details of Inquiries into child abuse scandals and in Court proceedings. Two examples are 1) comments in the judgement restoring Dr. Roy Meadow's standing, and 2) the general comments in the Goudge Report about why Dr. Charles Smith became the 'go-to guy' in North American paediatric forensic medicine despite being unqualified. These comments, and others I've seen in evidence before other inquiries, show doctors do, indeed, avoid Abuse cases because there's no money in them, the liability exposure is high, and reputational costs can be irreparable. 

One final comment: As we know, it wasn't Kempe who discovered unexplained broken bones in children, but radiographers working twenty years before. One question I've never seen addressed (with one exception) in the medical literature is why medicine, particularly radiography, was silent for so long? One might ask the same question about other forms of domestic violence and sexual intrusion. My own opinion - and it is just that - is that doctors (although not nurses) are generally poor collaborators. They appear to dislike losing control and therefore do not report abuse because they either do not trust other agencies, or they lack respect for those other agencies knowledge and skills base.

As I say, this is just my opinion, but a survey of this type might be a good opportunity to ask student medical professionals what they know about other agencies and how to collaborate with them. Do they know, for example, when they should avoid interviewing a child until there is a police presence so testimony is not contaminated? Are they aware of the necessary procedures to gather other evidence of a sexual or anatomical nature (photographs, swabs, etc.). What, if any, confidential patient information are they prepared to share with other agencies? Even if they strongly suspect neglect (which is almost always chronic), who would they call and why?

Such a line of questions would, hopefully do two things. First, it would help researchers to determine what students need to know (rather than the generic, 'they need to know more'), and second, it might encourage them to see themselves as part of a multidisciplinary team. 

In short, do professional medical students see themselves as part of a larger institutional matrix, or are do they see themselves as Lone Rangers responsible for, and capable of addressing, all their patients' needs. 

General Comments: The paper is well written and, from what I can tell, the statistical work is competent (although its not my field so don't take my word for it). I've mentioned the Munchausen's issue so, If I were you, I'd just drop the whole subject. If Munchausen's does exist, then its very rare and if it doesn't, this study doesn't need to attract a potentially politicized controversy. 

As you may have deduced, I come from a parallel discipline and my encounters with child abuse are obviously coloured by that discipline, but also the interactions I've had with the medical profession 'in the field' as it were. Thus, my belief that such a survey should not limit itself to pathological symptoms, but should also include knowledge of the entire complex of professions and disciplines that touch on child abuse and where medicine fits in. One instructive place to see this tension played out is in the Laming Report into the death of Victoria Climbie in the U.K. 

That said, I think this paper could serve as a warning beacon both in its identification of how little attention medical education pays to child abuse, but also in the apparent disinterest most practitioners have in the subject. I have my own ideas about why that disinterest exists, but they are only presumptions (although informed by experience) for the moment.

So, in my view, you need to amp up the volume a bit. I can understand why a doctor planning to make his living doing vanity plastic surgery might not care about child abuse, but any Family Doctor or paediatric consultant or paediatric pathologist should be thoroughly grounded in what is, after all, a part of their basic bread and butter. 

Author Response

  1. First, I'm puzzled why Munchausen's by Proxy is singled out as a specific category of abuse. It is generally accepted that it is an extremely rare condition and difficult to establish even when one finds it. I've read the original research by David Southall and Sir Roy Meadow, and while I'm sympathetic, one cannot avoid the political firestorms their various careers have created. I don't see how this is helpful to the present paper, and it could be harmful insofar as it could lead to sterile arguments about whether Munchausen's actually exists.  It certainly distracts from an important finding here, namely that while neglect is the most common form of abuse, it is the least likely to be recognized and understood. Now that I think about it, it would have been more useful to separate out Sudden Unexplained Death (or SIDS as was) since this is relatively common and highly problematic from a diagnostic point of view.

Thank you for comment. We have now added the following in the introduction section to clarify better for the readers the issue of Munchausen's by Proxy as following: Child maltreatment or CAN can be divided in the following main types: neglect, physical, psychological and sexual abuse. It should be highlighted that a particular form of child maltreatment, Munchausen by proxy syndrome, also known as medical child abuse or factitious disorder imposed on another, is a condition that involves characteristics of physical abuse, psychological maltreatment and medical neglect. [3, 4] Another form is the Munchausen by proxy syndrome, a relatively rare disorder of the primary caregiver subjecting the child to repeated medical interventions, based on a false or induced history.  [5] Munchausen syndrome by proxy has the T74.8 code in the tenth edition of the International Classification of Diseases (where T74 includes all known forms of child maltreatment). [6]

  1. Second, on page two the claim is made that health care professionals are the 'first to have contact with CAN'. This may be true for infants, but surely, for school age children, the more likely 'first' professionals are teachers? Moreover, would not one expect that police officers would be the first to see abuse connected with domestic violence? I think if this claim is to be made, then it needs to be substantiated. This also implicates the point early in the discussion that medical professionals may be reluctant to participate in child abuse investigations. A major problem with child abuse reports is the nature of the requirements to report. Neither in the methods, nor in the subsequent discussion section, is it clear what gets reported, by whom, and who does the follow-up work and generates the statistics. My jurisdiction is simple; it is mandatory for every citizen to report any suspicion of abuse to the police or child protection authority. It's not perfect, but it does have the virtue of clarity. The only exception is Solicitor/Client privilege. This doesn't seem to be the case for Greece, but it's not clear. Further, it seems the definition of abuse - or at least harm - is determined by the criminal code rather than by civil law as it is elsewhere. These matters need to be clarified since, at least in my jurisdiction, doctors are notorious for wanting to avoid the courts.

Thank you for this comment. Following your comment we have added the following:  Healthcare professionals, especially those working with children, play a pivotal role in addressing this complex pathology, since they are the first professionals coming in contact with the severe end of the spectrum especially during infancy. This is why they are required to be familiar with the physical and behavioral signs in children and caregivers indicative of CAN. They should also be aware of implementing the proper protocols and, most importantly, reporting to the relevant services. It is worth mentioning at this point that other professionals do come in contact with cases suspicious of CAN in the community such as teachers or policy officers. The latter highlights that the burden of the responsibility to recognize a sign of CAN is distributed amongst a wide spectrum of professionals as cases of CAN do not always end up admitted in the hospital. Moreover, it is of paramount importance for every citizen to be familiar with the process of reporting  any suspicion of abuse to the police or child protection authority.

  1. In passing I note there is a page numbering problem that needs to be addressed. This is now corrected

  1. Discussion: The first paragraph of the Discussion section states "In addition, the small participation rate may indicate that future healthcare professionals might be unwilling to deal with CAN in the future."  There is support for this view, but it's not necessarily in the academic literature. Rather, you will find it in the grainy details of Inquiries into child abuse scandals and in Court proceedings. Two examples are 1) comments in the judgement restoring Dr. Roy Meadow's standing, and 2) the general comments in the Goudge Report about why Dr. Charles Smith became the 'go-to guy' in North American paediatric forensic medicine despite being unqualified. These comments, and others I've seen in evidence before other inquiries, show doctors do, indeed, avoid Abuse cases because there's no money in them, the liability exposure is high, and reputational costs can be irreparable. As we know, it wasn't Kempe who discovered unexplained broken bones in children, but radiographers working twenty years before. One question I've never seen addressed (with one exception) in the medical literature is why medicine, particularly radiography, was silent for so long? One might ask the same question about other forms of domestic violence and sexual intrusion. My own opinion - and it is just that - is that doctors (although not nurses) are generally poor collaborators. They appear to dislike losing control and therefore do not report abuse because they either do not trust other agencies, or they lack respect for those other agencies knowledge and skills base.As I say, this is just my opinion, but a survey of this type might be a good opportunity to ask student medical professionals what they know about other agencies and how to collaborate with them. Do they know, for example, when they should avoid interviewing a child until there is a police presence so testimony is not contaminated? Are they aware of the necessary procedures to gather other evidence of a sexual or anatomical nature (photographs, swabs, etc.). What, if any, confidential patient information are they prepared to share with other agencies? Even if they strongly suspect neglect (which is almost always chronic), who would they call and why?Such a line of questions would, hopefully do two things. First, it would help researchers to determine what students need to know (rather than the generic, 'they need to know more'), and second, it might encourage them to see themselves as part of a multidisciplinary team. In short, do professional medical students see themselves as part of a larger institutional matrix, or are do they see themselves as Lone Rangers responsible for, and capable of addressing, all their patients' needs. 

Thank you for the above comments that summarize the current situation with the approach of CAN by many healthcare professionals. The infrastructure in each country is different and reporting services vary. The purpose of our article was to highlight the lack of structured training in the country towards CAN so that students understand the complexity of the issue very early on in their career and develop their skills and knowledge in the field. One of them should be the optimal interaction and cooperation with other disciplines and experts in the field. We have looked into that in the present study as we have focused on basic knowledge but this very important aspect may serve as a future goal for research targeting perhaps postgraduate training.

  1. General Comments: The paper is well written and, from what I can tell, the statistical work is competent (although its not my field so don't take my word for it). I've mentioned the Munchausen's issue so, If I were you, I'd just drop the whole subject. If Munchausen's does exist, then its very rare and if it doesn't, this study doesn't need to attract a potentially politicized controversy. As you may have dediced, I come from a parallel discipline and my encounters with child abuse are obviously coloured by that discipline, but also the interactions I've had with the medical profession 'in the field' as it were. Thus, my belief that such a survey should not limit itself to pathological symptoms, but should also include knowledge of the entire complex of professions and disciplines that touch on child abuse and where medicine fits in. One instructive place to see this tension played out is in the Laming Report into the death of Victoria Climbie in the U.K. That said, I think this paper could serve as a warning beacon both in its identification of how little attention medical education pays to child abuse, but also in the apparent disinterest most practitioners have in the subject. I have my own ideas about why that disinterest exists, but they are only presumptions (although informed by experience) for the moment. So, in my view, you need to amp up the volume a bit. I can understand why a doctor planning to make his living doing vanity plastic surgery might not care about child abuse, but any Family Doctor or paediatric consultant or paediatric pathologist should be thoroughly grounded in what is, after all, a part of their basic bread and butter. 

We would like to thank the reviewer for the overall positive feedback on our manuscript. Following your comments we have increased the volume of the manuscript highlighting issues raised by yourself and other reviewers of the present manuscript

Reviewer 2 Report

Dear authors,

Although the research was well-written in general, it would be more understandable for the reader to explain the curriculum in faculty of medicine and of nursing. The data from faculty of medicine are more than from nursing. Also the data from ‘working experience with children’ in ‘nonclinic’ are more. Do these make bias in answers? Student’s t-test could be used in Table 1, 2, and 3. Also the Discussion section is insufficient for practice. The reason of why the difference was NS in the ‘year of studies’ column of Table 4, and/or what suggestions could be given to improve the knowledge about CAN in faculties could be discussed in Discussion. 

Author Response

Dear authors,

  1. Although the research was well-written in general, it would be more understandable for the reader to explain the curriculum in faculty of medicine and of nursing. The data from faculty of medicine are more than from nursing.

Thank you for your positive comment on our manuscript. The curriculum in both medical and nursing faculties does not have a separate chapter on CAN. Each Faculty has a different approach on that which understandably affects the level of knowledge amongst students. As well noted by the reviewer we have more data from medical school rather than nursing school but this number reflects the number of students that voluntarily took part in the survey.

We have addressed the above issues as study limitations as follows:  Another study limitation is that the training varies among faculties and therefore that affects the level of knowledge amongst students. Moreover, more medical than nursing students participated in the survey as the completion of the questionnaire was done on a voluntarily basis

  1. Also the data from ‘working experience with children’ in ‘nonclinic’ are more. Do these make bias in answers?

The term working experience refers to whether students have had any kind of professional contact with children. Clinical experience refers to students that are doing or have completed their clinical practice on the pediatric ward round (such as final year students that are working as house officers or nursing students doing their practice on the ward) whereas the non clinical experience refers to students that have worked with children outside the healthcare setting, such as babysitting, camping activities etc. This classification may account for differences in answers but the level of student’s knowledge on CAN gradually builds up and exposure to cases eventually comes during their training or outside their training.

We have added the above clarification in the results section before Table 3

  1. Student’s t-test could be used in Table 1, 2, and 3.

We have now added the requested tests in Tables 1,2,3

  1. Also the Discussion section is insufficient for practice. The reason of why the difference was NS in the ‘year of studies’ column of Table 4, and/or what suggestions could be given to improve the knowledge about CAN in faculties could be discussed in Discussion.

Following your useful comment we have expanded the discussion before the limitation section:

Another finding of our study that is worth commenting on is that the years of studies did not seem to affect statistically significantly the level of knowledge. The latter is a worrying finding as one would expect that students in the final stages of their training are more aware of the issue. However, we did not demonstrate such a correlation. Therefore, faculties ought to take this into account and amend their curriculum to the extent that each healthcare professional in training obtains basic understanding of CAN that will enable safe practices for children in their future. In most faculties there is no separate curriculum on CAN but such training could be incorporated in the subject pediatric or forensic medicine or both. Another suggestion would be to adopt a stepwise approach meaning that training on CAN could be offered both in the early stages of training as well as in the final years. In addition, part of this structure training should be to build the understanding that all healthcare professional, irrespectively of their seniority and especially those that deal with children ought to early recognize signs of CAN and act appropriately. Of note, recent efforts have been made in the country to increase knowledge of medical students with the aid of novel methods such as  role-playing and  the use of manikins[29].

Round 2

Reviewer 2 Report

The article could be accepted in present form.